# Near-infrared observations of active asteroid (3200) Phaethon reveal no evidence for hydration

Driss Takir [1,7✉], Theodore Kareta [2], Joshua P. Emery[3], Josef Hanuš [4], Vishnu Reddy[2], Ellen S. Howell[2], Andrew S. Rivkin[5] & Tomoko Arai[6]

Asteroid (3200) Phaethon is an active near-Earth asteroid and the parent body of the Geminid Meteor Shower. Because of its small perihelion distance, Phaethon's surface reaches temperatures sufficient to destabilize hydrated materials. We conducted rotationally resolved spectroscopic observations of this asteroid, mostly covering the northern hemisphere and the equatorial region, beyond 2.5-μm to search for evidence of hydration on its surface. Here we show that the observed part of Phaethon does not exhibit the 3-μm hydrated mineral absorption (within 2σ). These observations suggest that Phaethon's modern activity is not due to volatile sublimation or devolatilization of phyllosilicates on its surface. It is possible that the observed part of Phaethon was originally hydrated and has since lost volatiles from its surface via dehydration, supporting its connection to the Pallas family, or it was formed from anhydrous material.

---

[1] JETS/ARES, NASA Johnson Space Center, Houston, TX 77058-3696, USA. [2] Lunar and Planetary Laboratory, University of Arizona, Tucson, AZ 85721-0092, USA. [3] Department of Astronomy and Planetary Sciences, Northern Arizona University, Flagstaff, AZ 86011, USA. [4] Institute of Astronomy, Charles University, CZ-18000 Prague 8, Czech Republic. [5] Johns Hopkins University Applied Physics Laboratory, Laurel, MD 20273, USA. [6] Planetary Exploration Research Center, Chiba Institute of Technology, Narashino, Japan. [7] Visiting astronomer at the Infrared Telescope Facility under contract from the National Aeronautics and Space Administration, which is operated by the University of Hawaii, Mauna Kea, HI 96720, USA. ✉email: driss.takir@nasa.gov

Asteroid (3200) Phaethon is an Apollo-type near-Earth asteroid (NEA) that is thought to be the parent body of Geminid Meteor Stream[1]. While originally suggested to be inactive, Phaethon develops a small dust tail for a few days after its perihelion[2,3]. The origin of this dust tail could be due to the desiccation and thermal breakdown of the surface[2] or the last gasps of a comet-like sublimation-driven activity[4]. It is unlikely for water ice to survive and the dust to be ejected from Phaethon via gas drag from ice sublimation because this asteroid's small perihelion distance and high temperature make ice unstable on even very short timescales[3]. The Earth-crossing asteroid 2005 UD was found to be dynamically similar to Phaethon, and both asteroids are thought to be fragments generated by the breakup of a primitive precursor object[5]. Understanding the nature of Phaethon's activity would help us both age-date and better understand the nature of one of the most easily seen and massive meteor showers.

With its effective diameter of $5.1 \pm 0.2$ km[6] and equatorial diameter of 6.1 km[7], Phaethon is one of the largest potentially hazardous asteroids (PHAs). Published values of the geometric albedo derived for Phaethon range from 0.08 to 0.16[6,8–10]. The rotational period of Phaethon is $P \sim 3.604$ h (ref. [6]). Bus and Binzel[11] classified this asteroid as a B-type according to the Small Main-Belt Asteroid Spectroscopic Survey (SMASS) taxonomy, though it is an F-type in the older taxonomy of Tholen[12]. Other B-type asteroids (e.g., (2) Pallas, (101955) Bennu) are composed of hydrated minerals, as evidenced by a strong absorption near 3 μm[13,14]. However, Phaethon's surface can reach temperatures sufficient to destabilize hydrated materials because of its small perihelion distance. B-types are carbonaceous and primitive asteroids and many of them are located in the high-inclination Pallas family that includes the second-largest asteroid (2) Pallas[15]. Japan Aerospace Exploration Agency (JAXA)'s DESTINY+ (Demonstration and Experiment of Space Technology for INterplanetary voYage, Phaethon fLy-by and dUst Science) mission will conduct a high-speed fly-by of Phaethon in the mid-2020s[16], possibly followed by a fly-by of NEA 2005 UD, which is likely a break-up body from Phaethon. National Aeronautics and Space Administration (NASA)'s (Origins, Spectral Interpretation, Resource Identification, Security, Regolith Explorer (OSIRIS-REx)) mission has rendezvoused with the other B-type NEA (101955) Bennu[17], which is spectrally similar to Phaethon in the visible and near-infrared spectral range (0.5–2.5 μm)[14,18]. Bennu

with a lower geometric albedo of $0.044 \pm 0.002$[17], was found to be hydrated[14] and recently revealed to be weakly active[19].

Here we present rotationally resolved spectra of asteroid Phaethon beyond 2.5-μm, which reveal no evidence for hydration on the surface of this asteroid. Our results indicate that volatile sublimation and phyllosilicate devolatilization may not be the cause for Phaethon's modern activity. Our conclusion supports the connection of Phaethon and the Pallas family.

## Results

**Near-infrared observations of asteroid (3200) Phaethon.** The shape model solution from Hanuš et al.[20] in Fig. 1 illustrates the orientation of Phaethon during our observations as a function of rotation phase. Given the aspect angle of ~53° and that our observations covered the whole rotation phase, most of Phaethon's surface was theoretically sampled by our data. Only the region close to the southern pole was not seen, although the observed surface was dominated by the northern hemisphere and equatorial region.

Figure 2 shows the thermally corrected long-wavelength cross-dispersed (LXD) spectra (sets a–j) of Phaethon. The band depths at 2.90 and their uncertainties, shown in Table 1, were derived using the technique described below in the Methods section. Sets i and j were acquired at the end of the observing night with an airmass of 1.849 and 2.130, respectively. Therefore, their signal-to-noise ratio is lower than for the other sets.

## Discussion

LXD spectra of Phaethon revealed that Phaethon's spectra are consistent with being featureless at the 3-μm band, suggesting that the surface (from depth of a few tens of microns) of this asteroid is not hydrated. These results are not in agreement with the finding of Lazzarin et al.[21], who detected an absorption band around 0.43 micron on Phaethon and attributed it to hydrated minerals. Our observations covered almost the whole surface except the region close to the southern pole that corresponds to about 10% of the whole surface of Phaethon, so the presence of hydrated minerals close to the south pole cannot be excluded.

The geometry of these observations largely matches that of Kareta et al.[9], whose NIR observations (0.7–2.5 μm) were taken earlier in the same night. Kareta et al.[9] found minimal variation across the surface of Phaethon except for subtle curvature in the

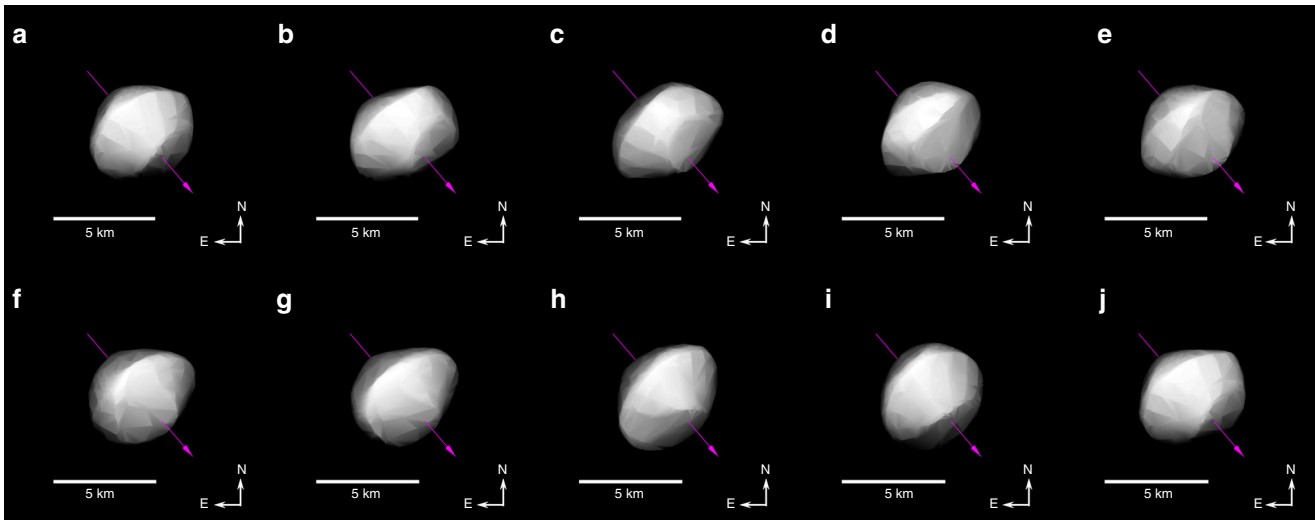

**Fig. 1 Rotation phases and orientation of (3200) Phaethon.** Our observations include 10 different rotation phases (**a–j**), mostly covering the northern hemisphere and equatorial region of Phaethon. The purple line indicates the position of the spin axis and the sense of the rotation.

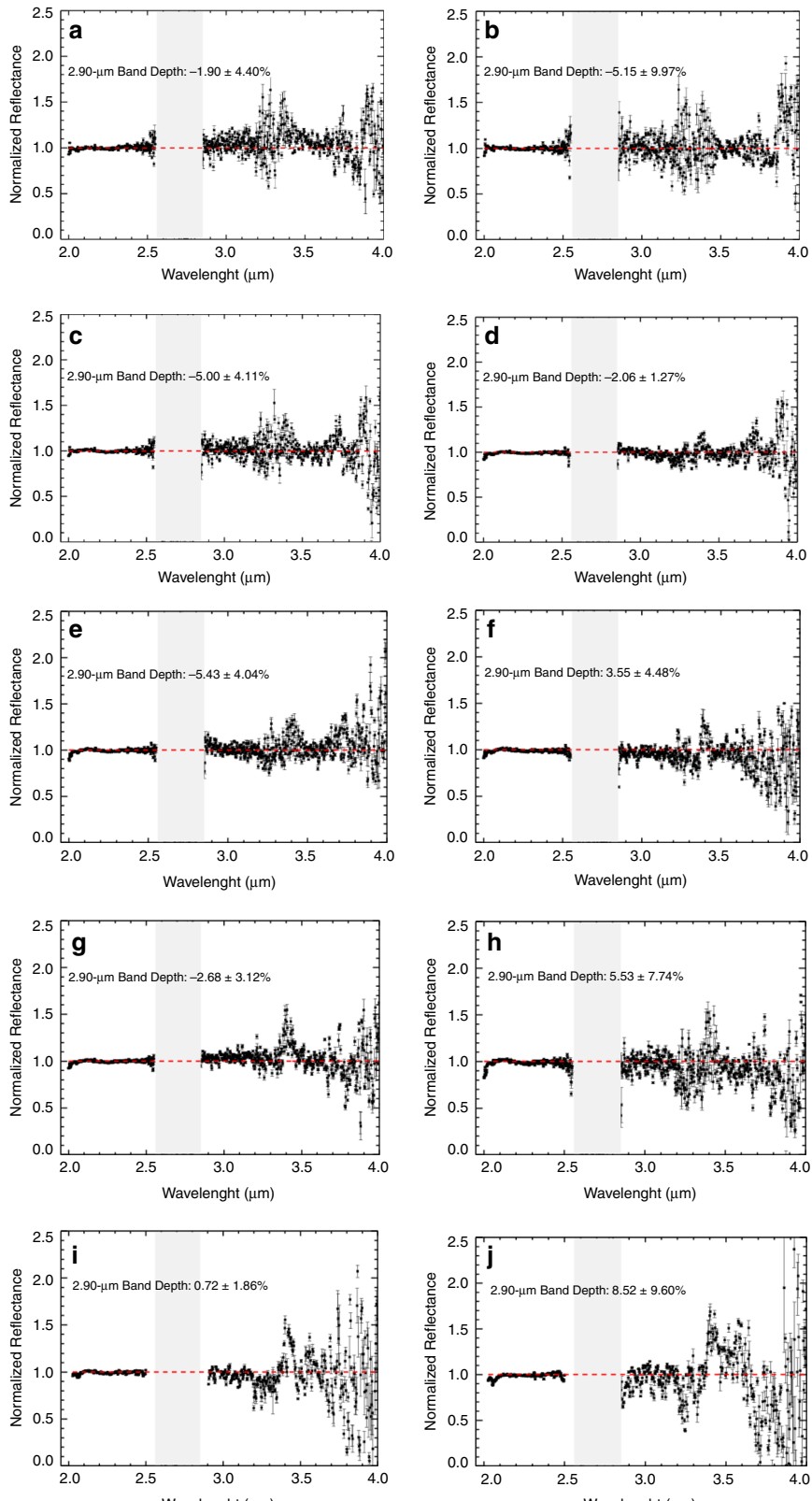

**Fig. 2 Thermally corrected normalized LXD spectra of asteroid Phaethon.** These spectra (**a**–**j**), which represent different sections of the asteroid, were found to be featureless at the 3-μm region, suggesting that the observed surface of this asteroid is not water-rich. The red dotted lines represent a continuum derived by setting the continuum reflectance at 2.45 μm to 1.0 and the slope to 0.00. The light gray bars (2.6–2.8 μm) mark wavelengths of strong absorption by water vapor in Earth's atmosphere. All spectra have been normalized to unity at 2.2 μm. Error bars were calculated using Spextool software and based on the Robust Weighted Mean algorithm with a clipping threshold of 8 (sigma).The value at each pixel is the weighted average of the good pixels and the uncertainty is given by the propagated variance.

**Table 1 Band depths at 2.90 μm with uncertainties for Phaethon.**

| Sets | Beaming parameter (η) | 2.90 μm band depth (%) | 1σ uncertainty (%) |
|------|-----------------------|------------------------|---------------------|
| Set a | 1.50 | −1.90 | 4.40 |
| Set b | 1.50 | −5.15 | 9.97 |
| Set c | 1.50 | −5.00 | 4.11 |
| Set d | 1.45 | −2.06 | 1.27 |
| Set e | 1.45 | −5.43 | 4.04 |
| Set f | 1.35 | 3.55 | 4.48 |
| Set g | 1.65 | −2.68 | 3.12 |
| Set h | 1.45 | 5.53 | 7.74 |
| Set i | 1.45 | 0.72 | 1.86 |
| Set j | 1.45 | 8.52 | 9.60 |

Asteroid Phaethon has no observable 3-μm band with >2σ detection at 2.90 μm. The beaming parameter values used in the thermal model to thermally correct each set are also included in this table.

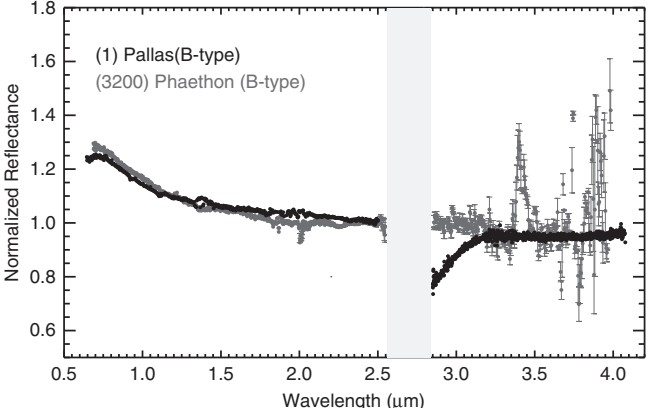

**Fig. 3 Spectra of B-type asteroids (2) Pallas and (3200) Phaethon (set g).** Both asteroids have blue-sloped spectra in the 0.5–2.5 μm spectral region. (2) Pallas has a prominent 3-μm band (~20%) suggesting its surface is phyllosilicate-rich unlike asteroid Phaethon. The prism spectrum of (2) Pallas is from the MIT-UH-IRTF Joint Campaign for NEO Reconnaissance database and the LXD spectrum of (2) Pallas is from Rivkin and DeMeo[13]. The prism spectrum of (3200) Phaethon is from Kareta et al.[9] Error bars were calculated using Spextool software and based on the Robust Weighted Mean algorithm with a clipping threshold of 8 (sigma). The value at each pixel is the weighted average of the good pixels and the uncertainty is given by the propagated variance.

NIR portion of its reflectance spectrum, perhaps related to differential heating over its surface. It is difficult to know whether surface heterogeneities might have existed on Phaethon prior to it being in its current orbit. Any original spectral heterogeneity may have been homogenized by the intense chemical and thermophysical changes brought on by its thermal environment. The numerical simulations of Hanuš et al.[6] showed that currently the equatorial region is facing the Sun during the perihelion passage, and thus is the most exposed region to the solar irradiation. However, due to dynamical effects, it was the southern hemisphere about 500 yr and 4 kyr ago that was facing the Sun during the perihelion. Finally, about 2 kyr ago, it was northern hemisphere's turn to be the most irradiated region. Clearly, the whole surface was exposed to similar environments during last few kyr.

Asteroid Phaethon loses ~$10^5$ kg per orbit assuming dust particles with an effective radius of ~1 μm (ref. [3]). While a small fraction of its total mass of ~$10^{14}$ kg (ref. [22]), this loss rate is more than enough to excavate some subsurface material, especially if the lost mass does not come from the whole surface uniformly. The fact that the observed portion of Phaethon's surface still looks as uniform as it does suggests that the current mass loss is possibly outpaced or matched by the processes (e.g., ballistic resurfacing) that serve to change and homogenize the surface, and thus that they act relatively quickly. If this interpretation is correct, one might expect that all sungrazing objects are essentially spectrally homogenized within a few orbits. Another reason that we might expect Phaethon to be more varied is the complex, cratered shape of the object seen in radar[7]. Perhaps the material excavated by those craters was similar to that of the surface materials, and thus Phaethon has been extensively heated throughout, or perhaps the topography we see is old compared to the recent heating of the surface layers.

de León et al.[23] suggested a compositional and dynamical connection between Phaethon and the B-type asteroid (2) Pallas both with high inclinations, 23° and 35°, respectively. These two asteroids have similar spectral features in the 0.5–2.5 μm spectral region (measured with the prism mode of SpeX) with blue-sloped spectra[24]. Masiero et al.[10] using NEOWISE observations found Phaethon to have a geometric albedo of 0.16 ± 0.02, consistent with the geometric albedo of Pallas. Ali-Lagoa et al.[25] also showed that the albedo of the B-types in Pallas collisional family ($p_v$ ~ 0.14) is higher than the average albedo of non-Pallas family B-types ($p_v$ ~ 0.07). However, Pallas exhibits a sharp and deep 3-μm band (~20%), attributed to the presence of phyllosilicates on its surface[13], unlike asteroid Phaethon (Fig. 3). The peak surface temperature on Phaethon's surface is estimated to be above 1000 K[3,5],

which exceeds the dehydration temperatures for serpentines (antigorite, chrysotile, lizardite) that range from 900 to 1000 K (ref. [26]). Carbonaceous chondrites were also found to experience appreciable loss of some volatile-elements and dehydration at T > 500 K (ref. [27]).

Another B-type asteroid, Bennu was found to be hydrated[14] and active[19]. Possible mechanisms explaining the weak activity of Bennu include dehydration of phyllosilicates[28]. If Phaethon really came from the Pallas family, then it seems likely that Phaethon was not necessarily volatile-free when it first plunged through the solar corona and reached temperatures above 1000 K. Rapid dehydration of the surface could have led to early activity, but given the anhydrous state of the surface indicated by the spectra presented herein, this mechanism is unlikely to drive Phaethon's current activity. Some more exotic mechanism like thermal degradation of the surface materials or solar wind sweeping[3] would be more likely to drive the current activity.

On the other hand, it is also possible that Phaethon was never hydrated to begin with (e.g., from an anhydrous portion of the interior of Pallas, or if the connection to the Pallas family proved to be incorrect). Differentiating between a 'dry' origin for Phaethon and a 'wet' origin for Phaethon is currently difficult. The current homogenous dry surface could be the end product of tens of thousands of years of intense heating. The lack of hydration could be from an originally volatile-rich target drying up or Phaethon could have simply formed on a parent body without many volatiles. The recent revelation of Bennu as an active body[19,28], considering the two objects have many similarities, leads to many questions about whether or not they could have a common origin and evolution. The DESTINY + mission, when it visits (3200) Phaethon in the 2020s, might be the only way to know for sure whether asteroids (3200) Phaethon and Bennu are related or not.

We conducted a spectroscopic analysis of asteroid Phaethon, covering mostly its northern hemisphere and equatorial region, beyond 2.5-μm to search for hydration evidence on asteroid (3200) Phaethon. We measured 3-μm spectra, indicative of hydration, of Phaethon with the LXD mode of the SpeX

**Table 2 Observational circumstances of asteroid (3200) Phaethon.**

| Sets | Mid. UTC | Magnitude (V) | Airmass | Sub-Earth latitude (°) | Rotational phase |
|------|----------|---------------|---------|------------------------|------------------|
| Set a | 10:17 | 11.26 | 1.128 | 36.96 | 0.00 |
| Set b | 10:42 | 11.25 | 1.153 | 36.91 | 0.12 |
| Set c | 11:06 | 11.25 | 1.189 | 36.87 | 0.23 |
| Set d | 11:38 | 11.25 | 1.258 | 36.81 | 0.37 |
| Set e | 11:59 | 11.24 | 1.320 | 36.77 | 0.47 |
| Set f | 12:20 | 11.24 | 1.391 | 36.73 | 0.57 |
| Set g | 12:36 | 11.23 | 1.324 | 36.69 | 0.64 |
| Set h | 13:05 | 11.23 | 1.622 | 36.64 | 0.78 |
| Set i | 13:35 | 11.23 | 1.849 | 36.58 | 0.92 |
| Set j | 14:02 | 11.22 | 2.130 | 36.53 | 1.04 |

This asteroid was observed on 12 December 2017. The columns in this table are the observation set, Mid. UTC, V-magnitude, airmass, sub-Earth latitude, and rotation phase.

spectrograph/imager at the NASA Infrared Telescope Facility. Our spectral analysis revealed that Phaethon is lacking hydration in the observed part of the asteroid, suggesting that this asteroid may have lost its volatiles during its evolution or was an anhydrous object before it was injected into its current orbit. Phaethon may have experienced extreme devolatilization and dehydration given that its perihelion (0.14 AU) is so close to the Sun. This conclusion supports the linkage of asteroids Phaethon and Pallas. It is uncertain whether Phaethon was originally hydrated and has since lost volatiles from its surface and interior via dehydration or it was formed from anhydrous material.

## Methods

**Observational techniques and data reduction**. We obtained spectra of Phaethon with the long-wavelength cross dispersed (LXD: 1.9–4.2 μm) mode of the SpeX spectrograph/imager at the NASA Infrared Telescope Facility (IRTF)[29] (Table 2). We obtained 10 LXD data sets (a–j) during the night of 12 December 2017, mostly covering a full rotational phase of the northern hemisphere and the equatorial region of Phaethon.

For Phaethon observations, we followed the same technique used in Takir et al.[30,31]. The spectral image frames were divided by a flat field frame measured using an internal integrating sphere. To correct for the contributions of OH line emission and the thermal emission from the sky (longward of ~2.3 μm), we subtracted spectral image frames of Phaethon and the solar analog standard star SAO 39985 (a G-type star close to Phaethon on the sky at similar airmass) at beam position A from spectral image frames at beam B of the telescope. After this subtraction, residual background was removed by subtracting the median background outside of data aperture for each channel. Spectra were extracted by summing the flux at each channel within a user-defined aperture. Asteroid spectra were divided by spectra of the solar analog measured close in airmass in order to remove telluric absorptions (mostly water vapor at these wavelengths). Wavelength calibration was conducted at $\lambda < 2.5$ μm using argon lines measured with the internal calibration box and at $\lambda > 2.5$ μm using telluric absorption lines. We processed Phaethon's LXD spectra using the Interactive Data Language (IDL)-based spectral reduction tool Spextool (v4.0)[32] provided by the NASA IRTF.

**Thermal excess removal in the 3 μm region**. The collected signal of Phaethon longward of 2.5 μm contains both thermal and reflected components (Fig. 4). Hence, we had to remove the thermal excess using the methodology described in Rivkin et al.[33] and references therein. To constrain Phaethon's model thermal flux, we used the Near-Earth Asteroid Thermal Model (NEATM)[34] which is based on the Standard Thermal Model (STM) of Lebofsky[35]. The measured thermal excess was fitted with a model thermal excess. Then, this model was subtracted from the measured thermal flux relative spectrum of Phaethon. In the thermal model, we used a visible geometric albedo of $p_v = 0.122 \pm 0.008$, derived by Hanuš et al.[6], and a slope parameter of $G = 0.06$, derived by Ansdell et al.[36] for Phaethon. The beaming parameter ($\eta$) is used in the thermal model to adjust the surface temperature to match the measured thermal flux (e.g., Harris and Lagerros[37]). We varied the values of the beaming parameters (from $\eta = 1.35$ to $1.65$) (Table 1) until we got the best thermal model for each set while keeping the geometric albedo constant at 0.122. When doing thermal tail corrections, the choice of the albedo does not matter too much because in most cases beaming parameter and albedo are degenerate, such that different pairs give exactly the same thermal tail. Both bolometric and spectral emissivities were assumed to be 0.9. Since we used the V-band albedo in the thermal model, we applied a K to V scale for Phaethon (K/V ~ 0.7)[23] to reconcile the two reflectance values at the two different wavelengths.

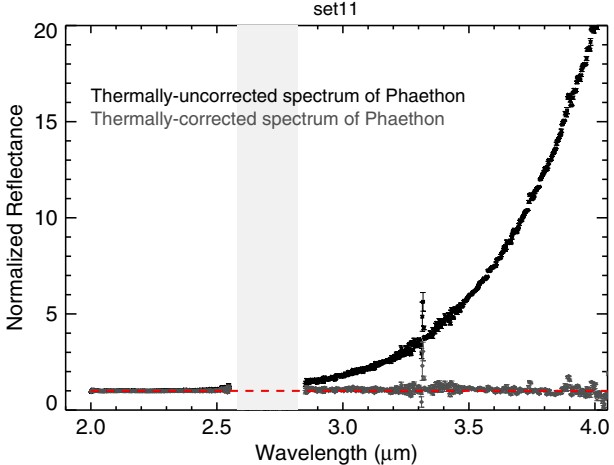

set11

**Fig. 4 Spectrum of (3200) Phaethon (set d) contains both thermal and reflected components longward of 2.5 μm (black).** The spectrum was corrected using the Near-Earth Asteroid Thermal Model (gray). The red dotted lines represent a continuum derived by setting the continuum reflectance at 2.45 μm to 1.00 and the slope to 0.00. The light gray bars (2.6–2.8 μm) mark wavelengths of strong absorption by water vapor in Earth's atmosphere. All spectra have been normalized to unity at 2.2 μm. Error bars were calculated using Spextool software based on the Robust Weighted Mean algorithm with a clipping threshold of 8 (sigma).The value at each pixel is the weighted average of the good pixels and the uncertainty is given by the propagated variance.

Asteroid Phaethon's spectra originally had a spectral resolution of $\lambda/\Delta\lambda = 2500$ and were binned by a factor of 9 to improve observational uncertainty.

**The 3-μm band depth and uncertainty**. Band depths at 2.90 μm were calculated using the following equation[38]:

$$D_{2.90} = \frac{R_c - R_{2.90}}{R_c} \qquad (1)$$

where $R_{2.90}$ is the reflectance at 2.90 μm, and $R_c$ is the reflectance of the continuum at 2.90 μm. The continuum was derived by setting the continuum reflectance at 2.45 μm to 1.0 and the slope to 0.00.

The uncertainty in $D_{2.90}$ is then[39]

$$\delta D_{2.90} = D_{2.90} * \sqrt{\left(\frac{\delta R_1}{R_1}\right)^2 + \left(\frac{\delta R_c}{R_c}\right)^2} \qquad (2)$$

where

$$R_1 = R_c - R_{2.90} \qquad (3)$$

and

$$\delta R_1 = \sqrt{(\delta R_c)^2 + (\delta R_{2.90})^2} \qquad (4)$$

$\delta R_c$ and $\delta R_{2.90}$ were derived using the uncertainty at each wavelength, calculated during the data reduction process.

## Data availability

We declare that data of asteroid (3200) Phaethon supporting this study's findings (Fig. 2) are available within the article and its Supplementary data file. In addition, raw data of Phaethon and standard stars used for processing are available from the authors upon reasonable request.

## Code availability

Spextool is an IDL-based spectral reduction program to reduce SpeX cross-dispersed and prism data. The code is available from: http://irtfweb.ifa.hawaii.edu/research/dr_resources/.

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

## Acknowledgements

D.T. was supported by NASA's Solar System Observations grant NNX17AJ24G. J.H. was supported by the grant 18-04514J of the Czech Science Foundation and by the Charles University Research program No. UNCE/SCI/023. V.R. and T.K. are supported by NASA Near-Earth Object Observations Grant NNX17AJ19G (PI: Reddy). J. P.E. was supported by NASA Near Earth Object Observations Grant NNX16AE91G. A.S.R. was supported by NASA Near Earth Object Observations Grant NNX14AL60G. We wish to thank NASA IRTF staff for their assistance with Phaethon observations. Spextool software is written and maintained by Mike Cushing at the University of Toledo, Bill Vacca at SOFIA, and Adwin Boogert at NASA InfraRed Telescope Facility (IRTF), Institute for Astronomy, University of Hawai'i. NASA IRTF is operated by the University of Hawai'i under contract NNH14CK55B with NASA.

## Author contributions

D.T. planned and conducted Phaethon's observations, reduced, processed, and analyzed the data, and wrote the first draft of the paper. T.K. contributed to the writing of the manuscript and helped shape the discussion section. J.P.E. contributed to the writing of the manuscript and provided feedback about the thermal excess modeling and correction analysis. J.H. contributed to the writing of the manuscript and provided the shape model analysis of Phaethon. V.R., E.S.H., A.S.R, and T.A. contributed to the writing of the manuscript and provided feedback. All authors reviewed and edited the manuscript.

## Competing interests

The authors declare no competing interests.
