## [Peer Review File · Nature Communications]

Reviewers' comments:

Reviewer #1 (Remarks to the Author):

This work presents, as far as I am aware of, the first collection of infrared (3-microns) spectra of Phaethon to detect the 2.7-2.9 microns absorption feature associated to the presence of hydrated silicates. This by itself deserves publication. The presented spectral data are used to shed some light into the origin of Phaethon, discussing two different scenarios: a 'dry' and a 'wet' origin. Although the lack of detection of such hydration feature is not as constraining as expected, and still the two scenarios are possible, I recommend this work for publication after some revision. I would like to see again the corrected manuscript before final acceptance.

General comments.

Methodology

- Line 81: I suggest to include here the value for Phaethon's rotational period.

- Lines 113-116: what is the reason for using the value of $p_v = 0.122$ for Phaethon's albedo, and not other measurements like the one reported by Kareta et al. (2018) ($p_v=0.08$) or the more recent measurement by Masiero et al. (2018), $p_v=0.16$? The same applies to the G value used by the authors: there are several publications in the literature providing lower values (even negative ones) for G in the case of Phaethon. Although I do not think this should be explicitly written in the main text, I would like to know what was the criterion of the authors to select such values of albedo and G slope for their modeling, and to actually see how changing these parameters changes their results.

- Line 119: I think the reader might wonder about the meaning of this K/V ratio. Please add a couple of sentences to explain why you need this value in your model.

Results

This section is composed of one single paragraph. I suggest to integrate it into the Discussion section, changing it to something like "Results and Discussion".

- Line 124: the authors refer here to thermally-corrected spectra (blue) and thermally-uncorrected spectra (black) in Figure 2, while the plot only shows the thermally-corrected ones. Please correct this.

- How are the uncertainties associated to the depth values computed? Is this uncertainty related to the dispersion in the data points? This needs to be properly described.

- Figure 2 is a nice way of presenting all your data in a single plot, but it does not help visualize your results. I suggest the authors to either change it (maybe you can show half your sample in one plot and half in another, so you can zoom in a bit) or keep it and select one or two spectra (maybe the ones with highest band depths) and present them in a separate plot. In addition, the caption of this figures says that the spectra are thermally corrected and normalized....to what wavelength? 2.45 microns? Please add this to the caption.

Discussion

- Lines 167: regarding the connection between Phaethon and Pallas, it should be added here the similarity between the albedo of Phaethon and those of both Pallas itself and the members of the Pallas family. In this respect, it is interesting the work from Ali-Lagoa et al. (2013), where they show that the albedo of the B-types in the Pallas collisional family ($p_v = 0.14$) is higher than the average albedo of B-types ($p_v=0.07$). This comment directly connects to what has been stated by the authors in line 210, where they mention the mismatch between the albedo values of Phaethon and Pallas. Although Kareta et al. (2018) obtained a value of $p_v=0.08$ for Phaethon, a very recent publication by Masiero et al. (2018), using WISE data, provides a value of $p_v = 0.16$. This should at least be mentioned and considered by the authors in their discussion.

- Figure 3. Please indicate in the caption which spectrum of Phaethon is shown in the plot (Set #?)

- Lines 184-194. This paragraph is a bit confusing. The authors comment on the possibility of Phaethon having an anhydrous origin. Then, in line 189 they suddenly introduce Bennu:

"(...) However, Bennu is likely also from the New Polana family...". It seems to me that some preceding info/sentence is missing. Later they state (line 191) that Phaethon is from the New Polana family. Is this a working hypothesis or is based on a published work? If so, the corresponding reference should be provided.

Minor corrections:

- Table 1. Rotational phase has no units.
- Line 138: "The red dotted line represents a continuum..."
- Line 167: "... de Leon et al. (2010)..."
- Line 172: "The surface temperature of Phaethon is..."
- Line 184: "Unlike Phaethon, Pallas..."

Reviewer #2 (Remarks to the Author):

Review of "Surface of Active Asteroid (3200) Phaethon" by Takir et al

General Comments:

This paper presents a solid observation, making the case that the surface of Phaethon shows no 3 micron hydration band. Unfortunately, the observation is not matched by a comparably good analysis. After a lot of words amounting to about half the paper (lines 142 - 229) the final sentence states the conclusion succinctly:

"It is unclear whether Phaethon was originally hydrated and has since lost volatiles from its surface and interior via dehydration or it was formed from anhydrous material."

Specific Comments:

Page 2: The perihelion activity cannot be due to buried ice because it would need to be within a thermal skin depth of the surface in order to respond at perihelion, as observed, but the temperature at this depth is much too high for ice to survive. Deep ice could exist but would not respond to perihelion heating because of the long conduction time.

The panels of Figure 1 all look very similar - what is the reader meant to see by comparing the panels? The figure is not effective.

Table 2 and Figure 2, compared:

The text states "Phaethon has no observable 3 μm band with $>2\sigma$ detection at 2.90 μm ". This statement is supported by Table 2, except for Set 4, which is $>3\sigma$ significant. But comparison with Figure 2 seems to contradict the Table. Specifically, $>3\sigma$ detection in Set 4 is not evident at all in Figure 2, while spectra from Sets 9 and 10 show clear dips at 2.9 microns relative to nearby continuum and yet these Sets are given 1 sigma errors in Table 2. I don't see the connection between the uncertainties in Table 2 and the spectra in Figure 2. Could that be explained a little in the text?

Figure 2 caption refers to a "gray band". What gray band?

Page 5 Pheathon is mis-spelled

Page 6 "the surface of this asteroid is not hydrated". Here is a good place to state to what depth the dehydration extends. It's a few tens of microns, isn't it?

Page 7: The fact that the surface still looks as uniform as it does suggests that the current mass loss is being outpaced or matched by the processes that serve to change and homogenize the surface, and thus that they act relatively quickly.

The natural process is ballistic resurfacing, as on comets.

Page 8 line 184: Some words are missing from the start of the sentence, or is it just not capitalized?

Page 8 Ohtsuk - no such person

Page 8 "the Earth's crossing asteroid". What?

Page 9: "Phaethon may experienced" - may have

Reviewer #3 (Remarks to the Author):

The paper deals with new interesting spectroscopic data of Phaethon. Phaethon has been studied during its several passages close to the Earth, in particular during the 2017 campaign, and especially in the visible region. Only very few information is available on its NIR spectrum. So these data deserve to be published.

My main comment is that the authors claim their results as definitive, that is, Phaethon is an anhydrous object, as clearly reported in the title. Instead I think that their results refer to a portion of the surface of Phaethon and all the paper should be written keeping this in mind.

After that, I think the paper should be published. It is important also as support of the Destiny mission.

Please find below my comments and some small corrections:

Page 2 lines 63-64: "B-types are carbonaceous and primitive asteroids and many of them are located in the high-inclination Pallas", please include a reference.

Page 4 lines 103-104: "we subtracted spectra of Phaethon and the standard star SAO 39985 at beam position A from spectra at beam B." The authors should give some more details of the reduction procedure.

Page 5 line 124: "shows the thermally-corrected LXD spectra (Sets 1 to 10) of Phaethon (blue)" I do not see the blue...

Page 5 line 125: 2.90...micron is missing

Page 5 line 126-127: "and slope of 0.00 are shown in Table 2" between 0.00 and are shown there should be "and they are shown"

Page 5 line 127: Pheathon is Phaethon

Page 6 line 143-144: "This study revealed that Phaethon is featureless at the 3- μ m band, suggesting that the surface of this asteroid is not hydrated" I partially agree with this conclusion as the authors didn't study the complete surface of Phaethon. It is probable that the surface of Phaethon is not hydrated owing to the strong heating, but this specific investigation is not enough to make a final conclusion on the whole surface. It could be better to say that it is not hydrated on the investigated portion of the surface.

Page 6 line 146-151: As before, even if it could be true that the whole surface of Phaethon is quite uniform, it cannot be excluded that some variations could be present, see for example Lazzarin et al. 2019 PSS. In this paper the authors find also an absorption band around 0.43 micron, typical of hydrated minerals.

Page 6 line 152-153: include this sentence "while a small fraction of its total mass of $\sim 10^{14}$ kg (Li and Jewitt 2013)" between commas.

Page 7 line 155: "The fact that the surface still looks as uniform as it does"the same as above, the authors have observed only a fraction of the surface of Phaethon and they generalize the uniformity found to the whole surface. I do not agree with this statement.

Page 8 line 184: "unlike Phaethon" unlike should have U capital letter.

Page 8 lines 188-190: "...mechanism like thermal degradation of the surface materials or solar wind sweeping (Jewitt 189 and Li, 2010). However, Bennu is likely also from the New Polana family and was found to be hydrated..." I do not like how the phrase However Bennu...starts. There not connection with the previous one, it should be rewritten better.

Page 8 lines 195-199: “A volatile-rich past for Phaethon would by necessity imply a period of intense devolatilization upon its injection into its current orbit, which would provide an easy explanation for the creation of the Geminids. Jewitt and Hsieh (2006) suggested that the Earth’s crossing asteroid 2005 UD is dynamically similar to Phaethon and that both are thought to be fragments generated by the breakup of a primitive precursor object. The scenario of Yu et al.....”

The authors should explain better the introduction of the similarity between Phaethon and 2005 UD in this context. They explain the consequence of the possible similarity with Polana and Pallas, but they do not explain that one with 2005 UD.

Last, I think that the title should be less categorical, for example it could be written as a question: Is the surface of Phaethon totally dehydrated? Or something similar..

Reviewer #1

This work presents, as far as I am aware of, the first collection of infrared (3-microns) spectra of Phaethon to detect the 2.7-2.9 microns absorption feature associated to the presence of hydrated silicates. This by itself deserves publication. The presented spectral data are used to shed some light into the origin of Phaethon, discussing two different scenarios: a 'dry' and a 'wet' origin. Although the lack of detection of such hydration feature is not as constraining as expected, and still the two scenarios are possible, I recommend this work for publication after some revision. I would like to see again the corrected manuscript before final acceptance.

Thanks. We appreciate your perceptive comments and suggestions.

General comments.

Methodology

- Line 81: I suggest to include here the value for Phaethon's rotational period.

Line 71: We added the value for Phaethon's rotational period.

- Lines 113-116: what is the reason for using the value of $p_v = 0.122$ for Phaethon's albedo, and not other measurements like the one reported by Kareta et al. (2018) ($p_v=0.08$) or the more recent measurement by Masiero et al. (2018), $p_v=0.16$? The same applies to the G value used by the authors: there are several publications in the literature providing lower values (even negative ones) for G in the case of Phaethon. Although I do not think this should be explicitly written in the main text, I would like to know what was the criterion of the authors to select such values of albedo and G slope for their modeling, and to actually see how changing these parameters changes their results.

Phaethon's geometric albedo of 0.08 derived by Kareta et al. (2018) using NIR spectra is lower than the original IRAS-derived albedo (0.11 ± 0.01 , Tedesco et al. 2002). The albedo derived by Masiero et al. (2019), using NEOWISE data, is higher. So in the thermal model we chose the albedo value of 0.122 ± 0.008 derived by Hanuš et al. (2016) (using Spitzer data and a thermophysical model utilizing the convex shape model) because it represents an average of those two values.

We agree with the reviewer that more appropriate G slope value for Phaethon needed to be used in the thermal modeling. In our thermal model (and in Line 129) we changed the G slope value from 0.15 to 0.06, which was derived by Ansdell et al. (2014). Overall, varying the G slope (and the geometric albedo) values in the thermal model does not affect our thermal correction of Phaethon spectra. After changing the G slope value, our results (especially the 2.90-micron band depth/ uncertainties) slightly changed, and our conclusions remained the same.

When doing thermal tail corrections, the choice of the albedo and G slope parameters (and other physical parameters) do not matter too much because in most cases those parameters are

degenerate, such that their multiple combinations gives exactly the same thermal tail and correction.

Line 133: We added this clarification.

- Line 119: I think the reader might wonder about the meaning of this K/V ratio. Please add a couple of sentences to explain why you need this value in your model.

Addressed.

Line 136: We added a few sentences explaining the K/V ratio and why we need that value in the model

Results

This section is composed of one single paragraph. I suggest to integrate it into the Discussion section, changing it to something like "Results and Discussion".

Addressed.

We added new text in the Results section; also, we moved some material from the methodology section to the results section (shape models) because we thought it fits better there.

- Line 124: the authors refer here to thermally-corrected spectra (blue) and thermally-uncorrected spectra (black) in Figure 2, while the plot only shows the thermally-corrected ones. Please correct this.

Addressed.

- How are the uncertainties associated to the depth values computed? Is this uncertainty related to the dispersion in the data points? This needs to be properly described.

Addressed.

Line 148: We added a more detailed description on how the uncertainties were calculated for Phaethon spectra.

- Figure 2 is a nice way of presenting all your data in a single plot, but it does not help visualize your results. I suggest the authors to either change it (maybe you can show half your sample in one plot and half in another, so you can zoom in a bit) or keep it and select one or two spectra (maybe the ones with highest band depths) and present them in a separate plot. In addition, the caption of this figures says that the spectra are thermally corrected and normalized...to what wavelength? 2.45 microns? Please add this to the caption.

Addressed.

Line 186: Instead of a single figure with 10 plots, we included 10 Figures with much a narrower range on the y-axis, one Figure/plot for each observed set. We also added in the caption that all spectra were normalized to unity at 2.20 microns.

I also added Figure 1 that shows thermally corrected and uncorrected spectra of Phaethon with a broader range on the y-axis.

Discussion

- Lines 167: regarding the connection between Phaethon and Pallas, it should be added here the similarity between the albedo of Phaethon and those of both Pallas itself and the members of the Pallas family. In this respect, it is interesting the work from Ali-Lagoa et al. (2013), where

they show that the albedo of the B-types in the Pallas collisional family ($p_v = 0.14$) is higher than the average albedo of B-types ($p_v=0.07$). This comment directly connects to what has been stated by the authors in line 210, where they mention the mismatch between the albedo values of Phaethon and Pallas. Although Kareta et al. (2018) obtained a value of $p_v=0.08$ for Phaethon, a very recent publication by Masiero et al. (2018), using WISE data, provides a value of $p_v = 0.16$. This should at least be mentioned and considered by the authors in their discussion.

We thank the reviewer for this comment.

Lines 68 and 240: We added those findings in the Introduction and Discussion sections.

- Figure 3. Please indicate in the caption which spectrum of Phaethon is shown in the plot (Set #?)

Line 253: We added Set #7 to the caption.

- Lines 184-194. This paragraph is a bit confusing. The authors comment on the possibility of Phaethon having an anhydrous origin. Then, in line 189 they suddenly introduce Bennu:

"(...) However, Bennu is likely also from the New Polana family...". It seems to me that some preceding info/sentence is missing. Later they state (line 191) that Phaethon is from the New Polana family. Is this a working hypothesis or is based on a published work? If so, the corresponding reference should be provided.

We improved the clarity of the discussion. In the discussion section, we focused on the Phaethon and Pallas connection, which is supported by our spectra and conclusion.

Minor corrections:

- Table 1. Rotational phase has no units.

Addressed.

- Line 138: "The red dotted line represents a continuum..."

Addressed.

- Line 167: "... de Leon et al. (2010)..."

Addressed.

- Line 172: "The surface temperature of Phaethon is..."

Addressed.

- Line 184: "Unlike Phaethon, Pallas..."

Addressed.

Reviewer #2

Review of "Surface of Active Asteroid (3200) Phaethon" by Takir et al

General Comments:

This paper presents a solid observation, making the case that the surface of Phaethon shows no 3 micron hydration band. Unfortunately, the observation is not matched by a comparably good analysis. After a lot of words amounting to about half the paper (lines 142 - 229) the final sentence states the conclusion succinctly:

“It is unclear whether Phaethon was originally hydrated and has since lost volatiles from its surface and interior via dehydration or it was formed from anhydrous material.”

We appreciate your comments. We have modified and improved the discussion section and added more supporting analysis elements in the methodology section.

Specific Comments:

Page 2: The perihelion activity cannot be due to buried ice because it would need to be within a thermal skin depth of the surface in order to respond at perihelion, as observed, but the temperature at this depth is much too high for ice to survive. Deep ice could exist but would not respond to perihelion heating because of the long conduction time.

Thanks for this comment.

Line 58: We added this clarification.

The panels of Figure 1 all look very similar - what is the reader meant to see by comparing the panels? The figure is not effective.

Figure 2 (Figure 1 in the older paper version) effectively shows the orientation and geometry of Phaethon throughout its rotation. So we disagree with the reviewer that it is not effective.

However, to make the Figure more effective and important in the paper, we added more info to the figure caption.

Table 2 and Figure 2, compared:

The text states "Phaethon has no observable 3 μm band with $>2\sigma$ detection at 2.90 μm ". This statement is supported by Table 2, except for Set 4, which is $>3\sigma$ significant. But comparison with Figure 2 seems to contradict the Table.

Specifically, $>3\sigma$ detection in Set 4 is not evident at all in Figure 2, while spectra from Sets 9 and 10 show clear dips at 2.9 microns relative to nearby continuum and yet these Sets are given 1 sigma errors in Table 2.

I don't see the connection between the uncertainties in Table 2 and the spectra in Figure 2.

Could that be explained a little in the text?

We agree with the reviewer that the > 3 sig. is not evident in Figure 3 (Figure 2 in the older version). This was because of an artifact at $\sim 3.3\text{-}3.4$ microns that we fixed by re-reducing the data. Specifically, the step where we had to merge LXD's spectra orders. After fixing that artifact, our conclusion (no hydration feature) is supported by the calculated of the uncertainty in Set 4 as shown in Table 1.

Sets 9 and 10 were acquired at the end of the observing night with high airmasses reaching 1.849 to 2.130. So the SNR in these two sets is not as great as the other sets. We re-reduced these two sets and after discarding the noised frames with the lowest SNR, our final results has improved a little bit for these two sets.

Figure 2 caption refers to a "gray band". What gray band?

Addressed.

Page 5 Pheathon is mis-spelled

Addressed.

Page 6 "the surface of this asteroid is not hydrated". Here is a good place to state to what depth the dehydration extends. It's a few tens of microns, isn't it?

That is right, the probed surface depth by our NIR reflectance observations is about a few tens of microns. We added this information in Lines 47 and 205.

Page 7: The fact that the surface still looks as uniform as it does suggests that the current mass loss is being outpaced or matched by the processes that serve to change and homogenize the surface, and thus that they act relatively quickly.

The natural process is ballistic resurfacing, as on comets.

Thanks. We changed the sentence in Line 224 to reflect these two comments

Page 8 line 184: Some words are missing from the start of the sentence, or is it just not capitalized?

We fixed that in the discussion section.

Page 8 Ohtsuk - no such person

Addressed. Corrected the author name : "Ohtsuka"

Page 8 "the Earth's crossing asteroid". What?

Fixed the typo.

Page 9: "Phaethon may experienced" - may have

Addressed.

Reviewer #3

The paper deals with new interesting spectroscopic data of Phaethon. Phaethon has been studied during its several passages close to the Earth, in particular during the 2017 campaign, and especially in the visible region. Only very few information is available on its NIR spectrum. So these data deserve to be published.

My main comment is that the authors claim their results as definitive, that is, Phaethon is an anhydrous object, as clearly reported in the title. Instead I think that their results refer to a portion of the surface of Phaethon and all the paper should be written keeping this in mind.

After that, I think the paper should be published. It is important also as support of the Destiny mission.

Thanks so much. We appreciate your thoughtful and helpful comments. We change the title of the paper to reflect our conclusions.

Please find below my comments and some small corrections:

Page 2 lines 63-64: “B-types are carbonaceous and primitive asteroids and many of them are located in the high-inclination Pallas”, please include a reference.

Thanks. Line 74: We added Cellino et al. (2002) reference.

Page 4 lines 103-104: “we subtracted spectra of Phaethon and the standard star SAO 39985 at beam position A from spectra at beam B.” The authors should give some more details of the reduction procedure.

We added more info about data reduction procedure in the methodology section.

Page 5 line 124: “shows the thermally-corrected LXD spectra (Sets 1 to 10) of Phaethon (blue)” I do not see the blue...

We fixed that in Figure 3.

Page 5 line 125: 2.90...micron is missing

Addressed.

Page 5 line 126-127: “and slope of 0.00 are shown in Table 2” between 0.00 and are shown there should be “and they are shown”

Addressed.

Page 5 line 127: Pheathon is Phaethon

Addressed.

Page 6 line 143-144: “This study revealed that Phaethon is featureless at the 3- μ m band, suggesting that the surface of this asteroid is not hydrated” I partially agree with this conclusion as the authors didn’t study the complete surface of Phaethon. It is probable that the surface of Phaethon is not hydrated owing to the strong heating, but this specific investigation is not enough to make a final conclusion on the whole surface. It could be better to say that it is not hydrated on the investigated portion of the surface.

Given the aspect angle of $\sim 53^\circ$ and that our observations covered the whole rotation phase, about 70-90% of Phaethon's surface was theoretically sampled by our data. Only the region close to the southern pole was not seen, although the observed surface was dominated by the northern hemisphere and equatorial region.

We changed the text and title to reflect this comment, Phaethon wasn't completely observed.

Page 6 line 146-151: As before, even if it could be true that the whole surface of Phaethon is quite uniform, it cannot be excluded that some variations could be present, see for example Lazzarin et al. 2019 PSS. In this paper the authors find also an absorption band around 0.43 micron, typical of hydrated minerals.

We thank the reviewer for pointing out Lazzarin et al.'s work. We addressed this in Line 206.

Page 6 line 152-153: include this sentence "while a small fraction of its total mass of $\sim 10^{14}$ kg (Li and Jewitt 2013)" between commas.

Addressed.

Page 7 line 155: "The fact that the surface still looks as uniform as it does"the same as above, the authors have observed only a fraction of the surface of Phaethon and they generalize the uniformity found to the whole surface. I do not agree with this statement.

Addressed.

Page 8 line 184: "unlike Phaethon" unlike should have U capital letter.

Addressed.

Page 8 lines 188-190: "...mechanism like thermal degradation of the surface materials or solar wind sweeping (Jewitt 189 and Li, 2010). However, Bennu is likely also from the New Polana family and was found to be hydrated..." I do not like how the phrase However Bennu...starts. There not connection with the previous one, it should be rewritten better.

Addressed.

Page 8 lines 195-199: "A volatile-rich past for Phaethon would by necessity imply a period of intense devolatilization upon its injection into its current orbit, which would provide an easy explanation for the creation of the Geminids. Jewitt and Hsieh (2006) suggested that the Earth's crossing asteroid 2005 UD is dynamically similar to Phaethon and that both are thought to be fragments generated by the breakup of a primitive precursor object. The scenario of Yu et al...." The authors should explain better the introduction of the similarity between Phaethon and 2005 UD in this context. They explain the consequence of the possible similarity with Polana and Pallas, but they do not explain that one with 2005 UD.

After rewriting the discussion section, we realized that the mentioning of 2005 UD in there does not fit because we do not have LXD spectra of 2005 UD to try to connect it to Phaethon and Pallas. Therefore, we moved the part that mentions 2005 UD to the introduction/background section.

Last, I think that the title should be less categorical, for example it could be written as a question: Is the surface of Phaethon totally dehydrated? Or something similar..

Thanks for this suggestion. We changed the title to reflect the fact that we were not completely

seeing Phaethon.

REVIEWERS' COMMENTS:

Reviewer #1 (Remarks to the Author):

I have read the revised version of this manuscript and all the requested changes have been done. The authors also did reply to all the raised questions and addressed all the specific issues highlighted by this reviewer. In my opinion, the paper is ready so I recommend it for publication.

Reviewer #3 (Remarks to the Author):

The authors revised the paper taking into account all my suggestions. I have read it and now in my opinion it is ready for publication.

REVIEWERS' COMMENTS:

Reviewer #1 (Remarks to the Author):

“I have read the revised version of this manuscript and all the requested changes have been done. The authors also did reply to all the raised questions and addressed all the specific issues highlighted by this reviewer. In my opinion, the paper is ready so I recommend it for publication.”

Thanks for the taking the time to review our revised version of the paper.

Reviewer #2 (Remarks to the Author):

Reviewer #2 submitted only confidential remarks.

Reviewer #3 (Remarks to the Author):

“The authors revised the paper taking into account all my suggestions. I have read it and now in my opinion it is ready for publication.”

Thanks for the taking the time to review our revised version of the paper.